# Sleep and Athletic Performance: A Multidimensional Review of Physiological and Molecular Mechanisms

**DOI:** 10.3390/jcm14217606

**Published:** 2025-10-27

**Authors:** Franciszek Kaczmarek, Joanna Bartkowiak-Wieczorek, Monika Matecka, Karolina Jenczylik, Kinga Brzezińska, Paulina Gajniak, Sonia Marchwiak, Katarzyna Kaczmarek, Michał Nowak, Michał Kmiecik, Joanna Stężycka, Kamil Krzysztof Krupa, Edyta Mądry

**Affiliations:** 1Student Scientific Society, Poznan University of Medical Sciences, 61-701 Poznan, Poland; franciszek.kaczmarek@pm.me (F.K.); paulina.gajniak@interia.pl (P.G.); sonix03marchwiak@wp.pl (S.M.); mn.nowak@pm.me (M.N.); 88871@student.ump.edu.pl (J.S.); 88992@student.ump.edu.pl (K.K.K.); 2Physiology Department, Poznan University of Medical Sciences, 61-701 Poznan, Poland; emadry@ump.edu.pl; 3Department of Occupational Therapy, Poznan University of Medical Sciences, 61-701 Poznan, Poland; mmatecka@ump.edu.pl; 4Faculty of Psychology and Law, SWPS University, 61-719 Poznan, Poland; kjenczylik@swps.edu.pl (K.J.); kaczmarek.katarzyna.swps@pm.me (K.K.); 5University Clinical Hospital of Poznan, Poznan University of Medical Sciences, 61-701 Poznan, Poland; kinga.brzezinska@usk.poznan.pl

**Keywords:** physiology of sleep, sleep disorders, sleep quality, sleep architecture, sleep deprivation, circadian rhythms, athletic performance, regeneration

## Abstract

Sleep is a fundamental biological process in athletes, indispensable for tissue regeneration, exercise adaptation, and injury prevention. Disruptions in sleep architecture and duration have been consistently associated with diminished physical performance and adverse health outcomes, impairing muscular strength, power output, and endurance capacity, and concurrently compromising cognitive function. On a physiological level, insufficient sleep disrupts endocrine homeostasis, elevating cortisol levels and reducing anabolic hormones such as testosterone and growth hormone. At the molecular level, sleep loss promotes the upregulation of pro-apoptotic gene expression and exacerbates pro-inflammatory signalling pathways. Optimal sleep duration and quality represent a critical “regenerative window”, essential for enhancing athletic performance and safeguarding physiological resilience. Ensuring adequate sleep among athletes can be effectively achieved through educational, behavioural, and nutritional interventions outlined in this review.

## 1. Introduction

Sleep is a fundamental biological process for maintaining the integrity and adaptability of the organism. Adults typically sleep an average of 7–8 h per day, which constitutes approximately one-third of their lives, underscoring its crucial role in maintaining homeostasis. However, there are significant differences in sleep duration between genders, geographic regions, and age groups [1]. Further, qualitative and quantitative sleep disturbances are associated with an increased risk of cardiovascular, metabolic, and psychiatric disorders [2].

For athletes, sleep takes on special importance as a key element of regeneration, exercise adaptation, and injury prevention. During sleep, a range of physiological processes occur that support tissue repair and energy resource replenishment [3] as well as anti-inflammatory mediators [2].

Sleep architecture also provides optimal conditions for hormonal, neurological, and anabolic processes [4]. Sleep is not merely passive rest but a highly active and complex biological process during which the body carries out mechanisms essential for health and performance [1]. On a hormonal level, sleep regulates the balance between anabolic and catabolic hormones. During deep sleep (N3), there is a surge in growth hormone, testosterone, and IGF-1 secretion—hormones crucial for tissue repair, protein synthesis, and muscle growth. Even a single night without sleep can reduce testosterone levels by nearly one-quarter, while chronic sleep deprivation promotes catabolic dominance. At the same time, insufficient sleep disrupts cortisol regulation; in excess, cortisol accelerates protein breakdown and hinders recovery [4]. 

The brain also works actively during sleep to organise and process information. In NREM stage 2, procedural memory consolidation occurs, strengthening motor skills and athletic techniques. REM sleep, on the other hand, promotes the creation of new strategies and integration of information into broader patterns, enhancing creativity and tactical thinking. According to the synaptic homeostasis hypothesis, sleep allows the weakening of less relevant neural connections, creating space for new ones and maintaining learning capacity. Conversely, sleep deprivation reduces focus, slows reaction time, impairs decision-making, and weakens motor coordination—factors critical in sports performance. In addition, inadequate sleep promotes the accumulation of neurotoxic β-amyloid, further impairing brain function [1,2,3,4].

On a somatic level, sleep is the body’s most important recovery window. During deep sleep, muscle repair and protein synthesis intensify, and energy stores, including muscle glycogen, are replenished. Extended sleep deprivation significantly reduces the ability to restore glycogen, directly lowering endurance and limiting the capacity for high-intensity exercise [3]. Sleep also regulates the immune system—supporting anti-inflammatory cytokine activity—while its deficiency raises levels of inflammatory markers such as IL-6 and CRP, which hinder tissue repair and delay the return to peak performance [2].

Insufficient sleep leads to declines in strength, power, endurance, and cognitive function and is also linked to higher rates of injury in young athletes. Additionally, inadequate sleep negatively affects mental health and regenerative capacity [5].

At the cellular level, sleep modulates the immune and hormonal systems, as well as inflammatory cytokines—all of which are crucial for post-exercise regeneration and adaptation [4]. Recent evidence also indicates the involvement of epigenetic mechanisms (e.g., DNA methylation and microRNAs) in the regulation of adaptive processes at the molecular level [6]. As a result, high-quality and adequate sleep duration serves as a natural “regeneration window”, supporting performance growth, health protection, oxidative stress reduction, and improved training adaptation [7].

Frequent and prolonged travelling—often involving transitioning across time zones—is inherent in the lifestyle of professional athletes and presents negative consequences, including disruptions to the circadian rhythm, resulting in both reduced sleep quantity and quality. Sleep disturbances may also manifest in less conventional ways, including gastrointestinal dysfunction, impaired cognitive and executive functioning, memory deficits, and emotional dysregulation. Collectively, these factors compromise recovery and athletic performance.

We provide a comprehensive overview of the role of sleep in sport, with particular emphasis on its regenerative functions and underlying physiological, molecular, and epigenetic mechanisms. Selected sleep disorders and their assessment methods are discussed, aiming to present sleep not only as a passive state of rest but as a dynamic and essential contributor to athletic performance, overall health, and injury prevention.

## 2. Methodology

A narrative review was conducted using the PubMed/Medline and Cochrane databases. The following English terms and their combinations were used: athletes, sleep, sleep deprivation, sleep loss, sleep disorders, insomnia, exercise, sport, performance, nap, physiology, pathophysiology, anxiety, stress, treatment, pharmacological, and non-pharmacological.

The search process was conducted stepwise. Initially, broad queries were applied, such as “athletes AND sleep” (7728 results, including 3945 articles published in the last five years), followed by more specific combinations targeting physiological and therapeutic aspects and focusing on studies published in the last five years. For example: “athletes AND sleep AND physiology” (1259 results), “athletes AND sleep AND sleep deprivation” (190 results), and “athletes AND sleep AND stress” (694 results). Additional queries addressed treatment approaches, including pharmacological (157 results) and non-pharmacological (36 results).

Approximately 7700 publications were identified. After screening the titles and abstracts, 178 articles were selected for evaluation. Each manuscript was then critically assessed and classified into two main groups: review articles and experimental studies. Subsequently, reference lists of the most relevant works were also analysed, leading to the inclusion of additional papers.

We excluded studies exclusively addressing psychological interventions, research involving children or older adults, and papers not directly related to sleep in athletes. Given the breadth of the subject, a selective choice of factors affecting sleep was made. The final selection focused on literature concerning the physiology of sleep, the bidirectional relationship between sleep and physical activity, the influence of diet and stress, sleep disorders, pharmacological and non-pharmacological treatments, as well as sleep assessment tools.

We present a simple Flow Diagram (Figure 1) illustrating the process of literature identification, screening, and inclusion.

## 3. The Physiology of Sleep

Sleep constitutes a fundamental psychophysiological process indispensable for preserving physiological capacity, health, and overall well-being, especially within athletic populations. Sleep and athletic performance remain in a close, bidirectional relationship. On the one hand, adequate sleep duration and quality form the foundation of recovery, supporting muscle repair, hormonal regulation, and cognitive processes that underlie decision-making and motor coordination [1,2,3,4]. Even short-term sleep deprivation leads to reduced strength, power, and endurance, impaired reaction time, as well as disturbances in cognitive functions—all factors that determine athletic outcomes. Chronic sleep deficiency exacerbates catabolic processes through elevated cortisol levels and decreased testosterone and growth hormone concentrations, thereby limiting protein synthesis and muscle recovery capacity [4].

On the other hand, physical activity itself and high training intensity influence sleep. Moderate exercise can improve sleep depth and facilitate recovery, whereas excessive training loads, pre-competition stress, or frequent travel across time zones may lead to circadian rhythm disruptions, difficulties falling asleep, and sleep fragmentation [1,2,3,4]. In young athletes, insufficient sleep increases the risk of musculoskeletal injuries, while in professional athletes, it reduces training efficiency and prolongs the time required to restore peak performance [3,4].

The literature emphasises that extending sleep duration (“sleep extension”) improves technical precision, reaction time, shooting accuracy, and aerobic performance, while short naps can partly compensate for nocturnal sleep deficits and support recovery [2,3,4]. Consequently, sleep is considered a key “recovery window” for athletes, with its insufficiency counteracting the benefits of even the most carefully designed training program [3,4].

Sleep consists of two major phases: rapid eye movement (REM) and non-rapid eye movement (NREM) sleep. NREM is further divided into three distinct stages—N1, N2, and N3—each characterised by specific brainwave patterns, eye movements, and levels of muscle tone (Figure 1). During sleep, a healthy athlete can cycle through the sleep stages approximately 4 to 6 times, with each full cycle lasting approximately 90 min. Many factors influence the quality and structure of sleep, including age, mood disorders such as depression, traumatic brain injuries, medications, sporting activity and circadian rhythm disturbances. Detailed physiological processes involved in these changes are described under the concept of “pathophysiology” [8].

Human sleep is characterised by distinctive neural oscillations observable through an electroencephalogram (EEG), which demonstrates the various functional stages of sleep. During Stage N1, the transition from wakefulness is marked by the reduction in alpha activity and the emergence of theta waves (4–7 Hz), often accompanied by vertex sharp transients and slow eye movements. These low-frequency rhythms signal the brain to move away from external awareness while drifting towards deeper rest [9].

**Figure 1 jcm-14-07606-f001:**
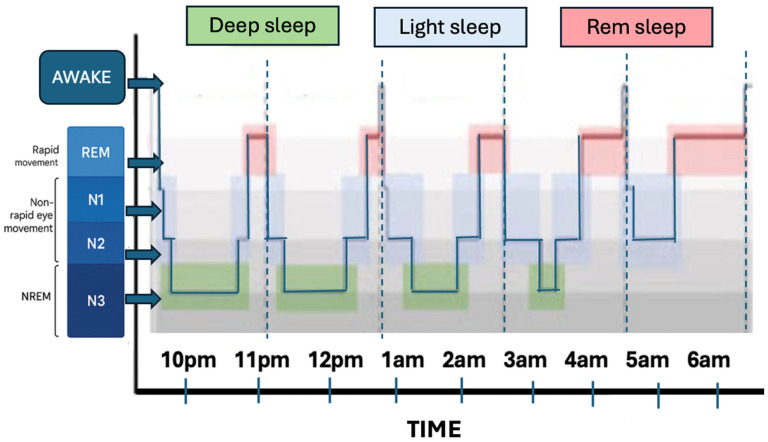
Physiological stages of sleep. Figure inspired by Driller et al. (2023) [10]. Created in BioRender. Kaczmarek, F. (2025) https://BioRender.com/heatk8t, (accessed on 20 October 2025).

Within Stage N2, two hallmark waveforms appear: sleep spindles and K-complexes. Sleep spindles consist of rhythmic bursts at 12–14 Hz lasting about 0.5–1.5 s, generated by thalamocortical circuits. They act as sensory gatekeepers and play a key role in memory consolidation, evidenced by increased spindle density correlating with better recall. K-complexes are large, biphasic waves exceeding 100 µV, serving to protect sleep from disruptions and aiding synaptic homeostasis. These too are linked to memory processes. There is strong evidence confirming a correlation between the duration of stage 2 sleep and the consolidation of motor memory. For athletes, sufficient time spent in stage 2 sleep is crucial for the long-term retention of techniques and motor patterns, which directly translates into subsequent performance [11].

Stage N3, or slow-wave sleep, is defined by delta waves (0.5–4 Hz, >75 µV), which reflect synchronised cortical activity facilitating physical and immune system restoration. Importantly, sharp-wave ripples, the high-frequency oscillations originating in the hippocampus, are tightly coupled with delta waves. These ripples mark moments when memories are replayed and transferred from the hippocampus to the cortex for long-term storage [12]. The deep sleep phase (N3) is particularly important, as this is when the mechanisms enabling muscle recovery and protein synthesis are amplified. Studies have demonstrated that insufficient slow-wave sleep disrupts growth hormone (GH) secretion and alters cortisol levels, thereby impairing post-exercise muscle recovery. Moreover, reduced slow-wave activity promotes inflammatory processes, as sleep restriction elevates pro-inflammatory cytokines (IL-6, CRP), which hinder muscle tissue repair and delay the restoration of optimal performance [4].

During REM sleep, the EEG exhibits mixed, low-amplitude theta (4–8 Hz) and beta/gamma (>30 Hz) activity together with REMs and muscle atonia. REM is further characterised by sawtooth waves, ponto-geniculo-occipital (PGO) waves, and synchronised theta–gamma oscillations. PGO waves, originating in the pons and propagating through the lateral geniculate nucleus to the visual cortex, align closely with hippocampal theta, supporting emotional memory processing and vivid dreaming. The coupling of theta phase and gamma amplitude—especially in the hippocampus—is linked to enhanced memory consolidation during REM [13] (Figure 2). In athletes, REM sleep facilitates the generation of new strategies and enhances the assimilation of information by strengthening memory traces associated with training.

Sleep architecture provides essential conditions for hormonal regulation, anabolic activity, and neurological processes. Although sleep may appear disadvantageous, because it increases vulnerability to predators, limits food intake, and restricts opportunities for reproduction, it persists throughout the animal kingdom. This persistence is likely explained by its crucial role in brain and behavioural development, as well as in memory reorganisation [4]. Three main hypotheses have been proposed to explain the role of sleep in the brain development of mammalian neonates. The Ontogenetic Hypothesis suggests that the amount of sleep or phasic sleep activity is closely related to brain development and maturation. According to the study by Frank, pharmacological deprivation of REM sleep can cause abnormal brain development or behaviour [14]. The studies performed on rodents have shown that it can cause changes in cholinergic and monoaminergic systems, neurotransmission, and the survival of neurons [15].

A second theory, the Consolidation Hypothesis, proposes that sleep provides a critical window for consolidating experience. This process occurs primarily during developmental periods of heightened, experience-dependent brain plasticity. The third explanation, the Synaptic Homeostasis Hypothesis (SHY), states that sleep enables the reduction in the number of synapses, selectively weakening less important connections while preserving those essential for transmitting critical information. This process makes room for the formation of new synapses [14]. Tononi et al., in their study, describe sleep as “the price the brain pays for its neuroplasticity”. Scientists agree with SHY and emphasise the role of normal sleep architecture in synaptic and cellular homeostasis, which must align with the plastic changes present during normal wakefulness. The balance of synaptic downscaling, learning processes, and synaptogenesis is necessary to prevent saturation of synaptic networks and maintain neuroplasticity [16]. The study by Colavito et al. indicates that sleep deprivation, when performed correctly, results in memory dysfunction in rodents. Although the study reports that sleep deprivation significantly impairs hippocampus-dependent spatial memory consolidation and contextual fear memory, the cued fear memory appears to remain unaffected by the loss of sleep, which shows that the effect of altered sleeping affects various types of memory differently [17]. At the synaptic level, although the total strength of connections underlying specific memories remains unknown, molecular, electrophysiological, and structural markers indicate a reduction in synaptic strength in the cerebral cortex and hippocampus. These findings are consistent with SHY, which proposes that synaptic downscaling during sleep is balanced by the selective strengthening of connections that support learning and memory reorganisation [18]. The duration and quality of sleep influence not only the neurological processes, but also the activity of the endocrine system. This is because concentrations of specific hormones, such as GH and cortisol, exhibit sleep-related variability, suggesting the close bidirectional connection between sleep and endocrine function. The study by Sassin et al. discovered that significant release of GH occurs during slow-wave sleep during stages 3 and 4 [19]. Additionally, the cortisol level rapidly grows about 30 min after awakening. This phenomenon is known as a cortisol awakening response. Wright et al. studied the relationship between sleep deprivation, circadian misalignment, and cortisol levels among other factors. In their study, they showed that acute sleep deprivation increased cortisol levels during the evening and early morning hours. On the other hand, chronic circadian misalignment decreased cortisol levels over the 24-h day [20]. Adequate sleep duration is necessary to maintain proper hunger and satiety. Spielberg et al. claimed that sleep restriction changed levels of appetite-related hormones. In their study, two consecutive nights of restricted sleep (four hours in bed) in young, healthy men resulted in a 24% increase in ghrelin (a hunger hormone) and an 18% decrease in leptin (a hunger-suppressing hormone). Moreover, participants demonstrated a stronger preference for calorie-dense foods rich in carbohydrates [21].

### 3.1. Neurochemical Mechanisms of Sleep and Wakefulness

The inhibitory neurotransmitter gamma-aminobutyric acid (GABA) plays a crucial role in initiating sleep by binding to GABA-A receptors in the brain [22]. Sleep-promoting neurons in the anterior hypothalamus release GABA, which suppresses wake-promoting regions in the hypothalamus and brainstem. GABA plays a central role in promoting slow-wave sleep, thereby supporting athletes’ physical recovery, hormonal balance, and muscle repair following intensive exercise. Adenosine also facilitates sleep by inhibiting orexinergic neurons in the hypothalamus while activating neurons in the preoptic area. Through its progressive accumulation during wakefulness, it promotes sleep pressure and facilitates recovery processes, making it a critical factor in athletes’ restoration of physical performance and prevention of overtraining [23,24].

Conversely, neurotransmitters such as acetylcholine (ACh), dopamine, norepinephrine, serotonin, histamine, and orexin peptides work together to maintain wakefulness. In athletes, these mechanisms are particularly important, as effective activation of the arousal system influences vigilance, reaction time, motor coordination, and physical performance. Regular physical training can modulate the functioning of these neurochemical pathways, leading to increased dopaminergic and noradrenergic activity and optimized orexin release, which supports better regulation of circadian rhythms, energy levels, and adaptive capacity to physical exertion. Acetylcholine release is highest during both wakefulness and REM sleep, but significantly lower during NREM [23]. Serotonin is produced by neurons in the raphe nuclei, norepinephrine by the locus coeruleus, and histamine by the tuberomammillary nucleus of the posterior hypothalamus. Orexin, synthesised in the hypothalamus, communicates with key arousal centres to sustain alertness [24].

Sleep is intertwined with the circadian rhythms that are linked to the daily rhythms of temperature and light. Throughout evolution, this linkage became more nuanced, but the basic rest-activity cycles still play a role in the life of organisms. Human sleep is a result of a series of molecules that are released locally, and when their effects are summed, they change the activity of the entire system [25]. These molecules play a significant role in keeping us asleep and awake. The level of dopamine, an important neurotransmitter that affects reward-motivated behaviour, increases during the night and is controlled by the circadian rhythms. The decrease in serotonin (5-HT), which regulates emotions and mental well-being, can disrupt the rest-activity cycle. Glutamate, on the other hand, sends information about the presence of light from the retino-hypothalamic tract (RHT) to the hypothalamic suprachiasmatic nucleus (SCN), which induces changes in sleep phases and increases during the day. Norepinephrine, which modulates attention, arousal, and cognition, increases during the night and stimulates the production of melatonin. Melatonin is a hormone produced by the pineal gland that regulates sleep onset and latency [26]. All of those substances and many more play a significant role in the process of sleeping. In athletes, a balanced regulation of these neurotransmitters and hormones is crucial for efficient recovery, maintenance of physical performance, and adaptation to training loads. Adenosine triphosphate (ATP), adenosine, nitric oxide (NO), prostaglandins, interleukin-1β (IL-1β), tumour necrosis factor α (TNF-α), brain-derived neurotrophic factor (BDNF), and nerve growth factor (NGF) all regulate sleep when released [25].

Sleep requirements and architecture vary significantly depending on age [Table 1]. During early life, sleep is longer and less consolidated, while in adulthood, it becomes shorter and more fragmented. Table 1 summarises these developmental changes in sleep patterns, circadian rhythms, and predominant sleep stages. Younger athletes may require longer sleep to support neurodevelopment, motor learning, and adaptation to increasing training loads. In adult athletes, shorter and more fragmented sleep can compromise recovery, cognitive function, and performance if not adequately managed [23,24,25].

### 3.2. Physiological and Molecular Benefits of Sleep

Adequate sleep has shown various advantages, including a regenerative effect on tissues and improvement in physical performance. Chen et al. investigated the association of sleep-disordered breathing and healing of diabetic foot ulcers (DFUs). The apnoea-hypoapnoea index (AHI) did not show a significant correlation with wound healing; however, total sleep time, sleep efficiency, and wakefulness after sleep onset (WASO) were associated with healing. Sleep fragmentation and hypoxemia predicted poor wound healing, high risk of ulcer recurrence, and even death in patients with DFUs [31]. Luo et al. suggested that circadian rhythms are intricately linked to bone reconstruction. The cycles of rest-activity affect bone energy metabolism by controlling the expression of the genes corresponding to dentin matrix protein 1 (Dmp1), osteopontin (Spp1), bone sialoprotein (Bsp), and osteocalcin (Bglap2), which participate in mineral deposition in the bones [32]. Elkhenany et al. observed how temperature changes, pH, carbon dioxide (CO2) level, NO, and hormone production affect stem cells during sleep. Specific levels of these parameters can stimulate growth, proliferation, differentiation, and self-renewal. Lung repair and regeneration are also affected by circadian rhythms [33].

Recently, the significance of sleep and its vital contribution to athletic performance, cognitive function, overall health, and psychological well-being have received growing recognition. Research consistently demonstrates that sleep restriction negatively affects physical performance in both submaximal and maximal efforts. Following sleep deprivation, physiological strain increases, as evidenced by elevated heart rate, ventilation, respiratory frequency, and lactate accumulation, which collectively lead to earlier onset of fatigue and reductions in VO_2_max. Sleep restriction also impairs anaerobic power, movement accuracy, muscle strength, and glycogen resynthesis, thereby limiting endurance and recovery capacity. Prolonged sleep deprivation further disrupts autonomic nervous system balance, potentially promoting overtraining-like states, whereas its effects on basic cardiorespiratory function appear less consistent and are likely dependent on the mode, intensity, and duration of exercise [34]. Acute sleep deprivation significantly impairs athletes’ sporting performance, with medium to large effect sizes depending on the type of performance and timing of assessment. The mechanisms underlying this decline include depletion of energy reserves, particularly muscle and liver glycogen, which limits the body’s ability to sustain and resynthesize energy during exercise. Cognitive impairments, such as reduced working memory, attention, and executive function, further compromise skill execution and decision-making. Additionally, sleep deprivation disrupts sleep architecture and increases inflammatory markers, negatively affecting muscle recovery and function. Performance decrements are most pronounced in high-intensity intermittent exercise, skill-based tasks, explosive power, and speed, with afternoon testing showing greater impairments than morning sessions. These effects are mediated by altered perception of fatigue, circadian rhythm disturbances, and sleep homeostasis, including adenosine accumulation, which collectively reduce neuromuscular efficiency, hormonal regulation, and overall physical capacity [35].

Adequate sleep and properly synchronised circadian rhythms enhance tissue repair, bone remodelling, and cellular regeneration in athletes by regulating the expression of genes involved in these regenerative processes, supporting recovery, adaptation to training, and peak performance. Conversely, sleep fragmentation, reduced sleep efficiency, or disrupted circadian cycles can impair these gene-mediated mechanisms, compromising recovery, diminishing performance, and increasing injury risk. Although all the above mechanisms of circadian rhythms and sleep can affect tissue regeneration and, consequently, the performance of athletes, it is essential to further research this area to draw firm correlations.

#### Molecular Consequences of Sleep Loss: Apoptosis and Circadian Disruption

The expression of genes associated with cell survival and death can be influenced by behavioural stressors like sleep deprivation and strenuous activity. In a study involving 20 healthy male participants, Norouzi Kamareh and colleagues found that 24-h sleep deprivation combined with anaerobic sprint testing significantly elevated pro-apoptotic markers: Bcl-2 associated X protein (BAX) and cell cycle and apoptosis regulator 2 (CCAR2), while simultaneously reducing anti-apoptotic: B-cell lymphoma protein 2 (BCL2) and the circadian regulator brain and muscle ARNT-like protein 1 (BMAL1) in peripheral blood mononuclear cells (PBMCs) (*p* < 0.05) [36]. This suggests that in athletes, particularly under conditions of sleep deprivation and high-intensity training, cellular stress and apoptotic signalling may be markedly increased, potentially impacting recovery and adaptation. BAX promotes mitochondrial outer membrane permeabilisation, an essential step in initiating apoptosis. CCAR2, meanwhile, intensifies p53-dependent apoptotic signalling, partly by downregulating BMAL1 through sirtuin 1 (SIRT1) inhibition [37]. Conversely, BCL2 stabilises mitochondrial membranes and inhibits apoptosis, while BMAL1 is known to support cellular rhythmicity and survival pathways [38].

L-arginine supplementation (1000 mg/day over eight weeks) appeared to reverse many of these effects. Treated individuals showed increased expression of BCL2 and BMAL1 and reduced levels of BAX and CCAR2 when compared to the effect of placebo controls [36]. This indicates that in athletic populations, L-arginine supplementation may help mitigate the pro-apoptotic effects associated with sleep deprivation and strenuous exercise, potentially supporting cellular recovery and adaptation. These observations are consistent with the capacity of L-arginine to modulate NO pathways and oxidative stress, both of which are connected to cell survival signalling [39].

Short-term sleep deprivation can lead to measurable cardiovascular strain. Yang et al. reported that subjects who remained awake for 24 h exhibited elevated resting blood pressure and an amplified pressor response to submaximal isometric exercise [40]. This was partly attributed to impaired baroreflex sensitivity and heightened sympathetic nervous system activation, as previously shown by Ogawa et al. [41].

Further evidence from Hori et al. indicated that a 2-h sleep during a deprivation period failed to mitigate these hemodynamic changes, pointing to the inadequacy of brief sleep recovery in resetting cardiovascular homeostasis [42]. Kato and Mullington also emphasised the role of sympathetic overdrive and inflammatory pathways in sleep loss–induced circulatory dysregulation [43,44]. For athletes, such responses may not only impair performance but may raise long-term risks for cardiovascular disease, particularly with repeated bouts of sleep deprivation.

Beyond systemic physiology, sleep loss exerts significant effects on multiple levels of human function, which is particularly relevant for athletes and individuals involved in high-intensity or physically demanding training. From a neurophysiological perspective, sleep loss can impair cognitive and motor function, which are critical for athletic performance. Controlled studies using positron emission tomography (PET) imaging have shown that even a single night of sleep deprivation leads to increased accumulation of β-amyloid in brain regions such as the hippocampus, parahippocampal gyrus, and thalamus, which in athletes may contribute to slower reaction times, impaired decision-making, and reduced coordination during training or competition [45,46]. Fragmented or insufficient sleep also disrupts connectivity between key neural regions, such as the prefrontal cortex and amygdala, affecting mood, attention, and stress regulation—factors directly impacting performance and recovery [47,48]. These neurobiological effects, observed even after short-term sleep loss, underscore the importance of uninterrupted, restorative sleep for maintaining optimal cognitive and motor function in physically active populations [49,50].

Acute sleep deprivation has been shown to shift gene expression toward pro-apoptotic pathways, increasing levels of BAX and CCAR2, while simultaneously reducing anti-apoptotic BCL2 and circadian regulator BMAL1 in peripheral blood mononuclear cells, potentially compromising cellular survival and rhythmicity [36]. Interventions such as L-arginine supplementation (1000 mg/day over eight weeks) have been shown to counteract some of the molecular stress effects induced by sleep deprivation, increasing BCL2 and BMAL1 expression and reducing BAX and CCAR2 levels compared to placebo, potentially supporting both cellular and neural resilience [36,39] (Figure 3).

From a cardiovascular standpoint, controlled 24-h sleep deprivation combined with submaximal isometric exercise elevates resting arterial blood pressure and exaggerates the pressor response to physical exertion. Even short recovery naps (e.g., 2 h) were insufficient to normalise these exaggerated hemodynamic responses, emphasising the critical importance of continuous restorative sleep for cardiovascular stability and athletic performance [42].

Altogether, these findings highlight that acute sleep deprivation, especially when combined with intense physical activity, can compromise cellular, cardiovascular, and neurophysiological systems in athletic populations. Increased β-amyloid accumulation in athletes due to insufficient sleep may further impair cognitive performance and recovery, making sufficient, high-quality sleep essential for maintaining peak performance, proper decision-making, and overall physiological resilience.

## 4. Methods and Tools for Assessing Sleep Quality and Duration

Research on sleep in athletes leverages advances in technologies, tools, and questionnaires designed to measure, monitor, and evaluate sleep in this population. Different methods of sleep assessment vary in terms of validity and reliability, and the choice of method depends largely on the specific purpose of monitoring. Objective measures, such as polysomnography (PSG) and actigraphy, provide quantitative data on sleep architecture, duration, efficiency, and fragmentation. PSG is considered the gold standard for evaluating detailed sleep stages but is resource-intensive and often limited to laboratory settings, while actigraphy allows for longitudinal, field-based monitoring, albeit with lower specificity for sleep stages. Subjective tools, including questionnaires like the Pittsburgh Sleep Quality Index (PSQI) or the Athlete Sleep Screening Questionnaire (ASSQ), capture perceived sleep quality and disturbances but may be influenced by individual bias or recall inaccuracies. Sleep trackers, including wearable and nearable devices, represent some of the most rapidly evolving tools in this field. Most studies focus on the effects of sleep restriction or deprivation, as well as the cumulative impact of sleep deficit, on various physiological functions and athletic performance. The use of heterogeneous assessment methods frequently results in divergent findings, complicating direct comparisons and synthesis. Consequently, interpreting the influence of dietary or lifestyle interventions on sleep in athletes requires careful consideration of the measurement approach, and conclusions are often contingent on whether objective or subjective metrics were employed [51,52].

Proper sleep is often defined using the Pittsburgh Sleep Quality Index (PSQI), developed by Buysse et al. (1989) [53] to assess sleep quality. It contains 19 questions covering quality, duration, latency, efficiency, disturbances, medication use, and daytime dysfunction. A PSQI score > 5 indicates poor sleep, while scores ≤ 5 indicate good sleep [53]. After 35 years of use, updates have been proposed, including the shorter PSQI-2 (De Menezes-Júnior et al., 2025) based on sleep duration and subjective quality (0–6 points, higher = worse sleep) [51].

Polysomnography (PSG) records sleep using a camera and multiple sensors, analysed by technicians. Reports include the AHI, with scores < 5/hour considered normal; higher values may suggest sleep apnoea [52]. Although PSG produces an accurate assessment of nocturnal sleep, it is invasive and less accessible. Actigraphy (ACG), using wearable devices to track movement and light over multiple days at home, is non-invasive but less precise as it relies solely on movement [54]; Zeiter et al. found increased discrepancies with PSG when wake time during a sleep episode is longer [55].

PSQI and ACG studies are most commonly used on athletes—both are non-invasive and easy to use in the home environment. Other tools include sleep diaries (e.g., Consensus Sleep Diary [56]), chronotype scales (MEQ, MCTQ), Insomnia Severity Index (ISI) [57], Athens Insomnia Scale (AIS) [58], and Athlete Sleep Behaviour Questionnaire (ASBQ) [59]. All require patient cooperation to ensure reliable results.

## 5. Sleep and Performance in Athletes: Selected External Factors

Sleep quality in athletes can be disrupted by various environmental factors—such as diet, transmeridian travel, and ambient conditions—that can significantly affect their recovery and performance.

Travelling across at least four time zones leads to a desynchronisation of the circadian rhythm, resulting in jet lag symptoms such as sleep disturbances, difficulty concentrating, irritability, mild depression, fatigue, and gastrointestinal problems. In athletes, this may lead to reduced physical and mental performance as well as an increased risk of injury [60]. The direction of travel plays a critical role in the adaptation process. East–west travel is generally associated with more adverse effects compared to north–south travel. A study analysing 20 years of data from Major League Baseball demonstrated that eastward travel exerted a significantly greater negative impact on winning percentages compared to westward travel, with this effect reaching statistical significance only for home teams, which experienced an approximate 3.5% decrease in wins (*p* < 0.05) [61]. For most individuals, eastward travel poses greater adaptation challenges than westward travel, though this is not universally the case. Eastward transitions require an advancement of the circadian rhythm, a process that is inherently more difficult, while westward travel induces a phase delay. In the latter case, competition times may coincide with the biological night of the athlete, significantly impairing physical and cognitive performance. While in the short term, jet lag may reduce the likelihood of optimal performance or victory, chronic travel across time zones, particularly in the east–west direction, acts as a potent physiological and psychological stressor. This is associated with increased risk of injury, cognitive impairments, and mood disturbances [61] (Figure 4).

The impact of circadian misalignment and associated sleep deficits on athletic performance and health varies across sports disciplines. For example, aerobic training requires greater energy expenditure than anaerobic training, and as such, endurance athletes generally exhibit a higher sleep need [62]. These observations underscore the necessity for a discipline-specific and individualised approach to training schedules and competition planning. Although circadian disruption affects athletic performance, its influence is often difficult to predict due to the interplay of multiple factors such as travel logistics, light exposure, and event timing.

To mitigate these adverse effects, the following recommendations have been proposed to enable circadian adaptation:(1)Pre-adaptation—gradually shift sleep timing by 30–60 min in the days preceding travel, combined with appropriate light exposure (morning light for eastward travel, evening light for westward travel),(2)Post-arrival adaptation—adjust sleep, light exposure, meals, and training schedules to the local time zone, avoiding bright light at inappropriate times,(3)Recovery—allow a minimum of 24 h rest following travel; naps lasting 20–90 min can support physical and cognitive performance,(4)Sleep hygiene—limit caffeine intake and screen exposure in the evening, utilise eye masks and earplugs, and avoid intensive activities immediately after waking,(5)Technological support and education—use apps for planning light exposure and caffeine use; provide education on sleep and circadian rhythms for athletes and support staff [63] (Figure 5).

Diet plays a significant role in the sleep quality of athletes. The most cited dietary factors associated with improved sleep in general populations include meal timing, macronutrient composition, and supplementation [64]. The timing and composition of meals act as key *zeitgebers* (external time cues, e.g., light or meal timing) for peripheral molecular oscillators, particularly within hepatic and gastrointestinal tissues. Disruptions in feeding rhythms can induce desynchronization between peripheral clocks and the central pacemaker in the suprachiasmatic nucleus, resulting in impaired glucose and lipid homeostasis, altered hormonal secretion, and heightened systemic inflammation. In athletes, such circadian misalignment may have significant consequences: it can negatively affect substrate metabolism during exercise, modulate endocrine responses (e.g., cortisol, testosterone, growth hormone), and disturb sleep architecture and recovery processes. Consequently, misalignment between feeding schedules and circadian rhythms may reduce exercise capacity, prolong recovery time, and increase susceptibility to injury and overtraining [65]. Consuming tryptophan-rich foods (such as turkey, fish, eggs, yoghurt, and nuts) and complex carbohydrates can enhance the production of melatonin—the hormone responsible for regulating sleep. Melatonin levels appear to be sensitive to dietary macronutrient composition, with protein and carbohydrate intake indirectly affecting its biosynthesis through modulation of the tryptophan to large neutral amino acid ratio. An increase in this ratio may enhance the availability of tryptophan, stimulate melatonin synthesis, and consequently improve sleep, which can be achieved through the consumption of carbohydrates or protein sources that are high in tryptophan and low in other large neutral amino acids [66]. The Mediterranean diet, rich in fibre, B vitamins, magnesium, and zinc, has been associated with improved sleep quality. In contrast, highly processed diets—high in simple carbohydrates and saturated fats—may negatively affect sleep quality by promoting inflammation [67]. The consumption of high glycemic index (GI) carbohydrates in the evening, particularly when consumed 2–4 h prior to bedtime, has been investigated for its effects on sleep in athletic populations. Several studies have reported that evening high-GI meals can reduce sleep onset latency, suggesting a potential facilitation of sleep initiation. However, total sleep duration does not appear to be significantly altered by such dietary interventions. Notably, evidence from a study involving state-level basketball players indicates that high-GI evening meals may negatively affect subjective sleep quality, highlighting the possibility of population-specific or context-dependent effects [68].

Caffeine consumption prior to or during competition is common in up to 90% of athletes, establishing it as the most widely used psychoactive substance. Moss et al. showed that higher caffeinated beverage intake was related to poorer sleep quality and increased the risk for disordered breathing while sleeping among endurance athletes. Individuals consuming 1.5 cups or fewer of caffeinated beverages per day reported better sleep quality, as measured by the Athlete Sleep Screening Questionnaire (ASSQ) global score, compared to those athletes with higher levels of caffeine consumption [69].

Athletes are particularly susceptible to ω-3 polyunsaturated fatty acid (PUFA) insufficiency, a factor that should be considered in the development of their nutritional strategies. The most effective method of ensuring adequate intake of ω-3 PUFAs is through the consumption of fatty fish, such as salmon, mackerel, trout, and sardines, which are primary dietary sources of eicosapentaenoic acid (EPA) and docosahexaenoic acid (DHA). Plant-based sources of alpha-linolenic acid (ALA) include flaxseeds and flaxseed oil, chia seeds, and walnuts. In cases where achieving sufficient dietary intake is difficult or impractical, appropriately dosed supplementation may be an effective approach. ω-3 PUFAs play a vital role in maintaining optimal physiological functioning, influencing both overall health and physical performance. Supplementation with ω-3 PUFAs—particularly EPA and DHA—has positively affected endurance capacity, muscle strength, recovery processes, and immune function. Additionally, ω-3 PUFA supplementation may enhance sleep quality [70]. Studies conducted on Syrian hamsters support the hypothesis that omega-3 deficiency may disrupt circadian rhythms through altered melatonin secretion. In the experimental group, the nocturnal peak of melatonin secretion was 52% lower than that in the control group. This reduction may explain the higher levels of locomotor activity during the day and night in animals with n-3 PUFA deficiency [71] (Figure 6). However, human studies investigating this relationship are currently lacking and are warranted to determine whether similar mechanisms are present in human populations.

Overall, patterns of nutrient intake and meal timing can influence sleep duration, efficiency, and the ease of sleep initiation. Certain evening dietary strategies, such as the consumption of specific macronutrients, may facilitate faster sleep onset, whereas other habits may be associated with lighter, more fragmented sleep. Emerging evidence suggests that sustained supplementation with probiotics may improve both objective and subjective measures of sleep, potentially by modulating gut–brain signaling and reducing systemic inflammation. Similarly, intake of functional foods such as tart cherry juice or beetroot juice shows preliminary promise in supporting sleep, likely through their effects on melatonin availability and nitric oxide–mediated vascular function, respectively [68]. Although these findings are promising, the evidence remains limited, and further research is needed to establish their efficacy.

### 5.1. Sleep Deprivation and Athletic Performance

Craven et al. completed a meta-analysis that highlighted the negative impact of sleep deprivation and late sleep restriction on the physical performance of athletes. Sleep loss was detrimental to the athletic performance both in the AM (ante meridiem) and PM (post meridiem), but the outcome was worse in the PM. Anaerobic power, speed, and power endurance, as well as high-intensity interval exercise, were all negatively affected by disturbances in the sleep cycle [72].

Sleep disorders change our physical performance. Hu et al. evaluated professional Chinese winter sport athletes and revealed that 13% of the participants scored high for symptoms of insomnia, while some demonstrated mental health problems [5]. Young athletes can also experience related problems. A study documented poor sleep quality (PSQI ≥ 5) concerns in 61.1% and 63.8% of the under-14 (U14) and under-17 (U17) participants, respectively, while 8.3% of the U14 and 8.4% of the U17 presented mild insomnia [73]. Craven et al. demonstrated that acute sleep loss, defined as ≤ 6 h of sleep within any 24 h, negatively affected strength, anaerobic power/capacity, endurance, and skill-based activities. Sleep loss had a significant (*p* = 0.001) impact on performance outcomes. Total sleep deprivation and late sleep restriction protocols were associated with the most pronounced declines across various exercise modalities. Meta-regression analyses revealed that exercise performance decreased by approximately 0.4% for every additional hour spent awake following sleep loss. Based on these findings, physical activity after sleep loss should be scheduled as early as possible to minimise performance impairments [72]. Brotherton et al. demonstrated that performance in maximal grip and bench press (reduction in average power, average force, and peak velocity), as well as submaximal leg press values (average power), were significantly lower compared to normative habitual sleep after partial sleep deprivation. It is worth noting that a 1-h post-lunch power nap (13:00) enabled performance values to return close to normal [74]. These results are in contrast with those of Gallagher et al., who found that implementing a 30—or 60-min nap at 1:00 PM after two consecutive nights of partial sleep restriction did not influence weightlifting performance but did lead to improvements in cognitive function [75]. The meta-analysis by Mesas et al. [76] demonstrated that a daytime nap lasting between 30 and <60 min had a positive effect on physical performance, including strength, endurance, and speed, following partial sleep deprivation, as well as after regular sleep. Moreover, physical performance improved progressively over time after the nap. Therefore, a minimum interval of 60 min post-nap may be necessary to reduce the negative effects of sleep inertia.

Additionally, a nap duration of 30 to <60 min was associated with improved cognitive function and reduced fatigue [76]. Napping affects not only athletes after sleep deprivation but also after normal nighttime sleep. In the meta-analysis by Boukhris et al., napping prior to afternoon training sessions showed a positive impact on performance outcomes. However, the final result may depend on nap durations, the time of day of naps, sleep inertia, and exercise type. Boukhris et al. showed that daytime napping improves performance in the 5-m shuttle run test, with no effect on muscle force. The lack of observed improvement may be related to differences in the exercises used to assess muscle force [77]. Blumert et al. suggested that 24 h of sleep loss does not affect weightlifting performance during a high-intensity training session during which athletes performed the snatch, clean, jerk, and front squat. Although no decrease in performance was observed, the athletes did experience fatigue, confusion, total mood disturbance, and sleepiness [78]. Gong et al. demonstrated that acute sleep deprivation significantly (*p* < 0.001) affects overall sporting performance, with a more pronounced decline in the afternoon compared to the morning. These findings are consistent with the results reported by Brotherton et al., who suggested that the extent of performance decline depended on the type of exercise and time of day. Specifically, the most significant impairments observed were in aerobic endurance and skill control indicators during the morning, and in explosive power, speed, and skill control indicators in the evening [35]. In the comprehensive meta-analysis conducted by Kong et al., the performance parameters most susceptible to the negative effects of sleep deprivation were skill control, aerobic endurance performance, speed performance, ratings of perceived exertion, explosive power, and maximal strength. The type of sleep deprivation affected specific performance parameters differentially. Furthermore, afternoon testing produced significantly poorer outcomes compared to those tested in the morning. It is worth noting that short-term sleep deprivation did not have a significant impact on anaerobic endurance performance, and athletes were found to be more sensitive to the effects of sleep deprivation than non-athletes [79].

Knufinke et al. suggested that a shorter sleep onset latency was associated with improved performance in tasks requiring gross motor skills (e.g., vertical jumps, maximal sprints, or constant power output tests). Furthermore, analysis of specific sleep stages indicated that spending less time in light sleep was linked to enhanced motor performance, whereas time spent in deep sleep and REM sleep was not significantly associated with gross motor skill outcomes [80].

The results of sleep interventions in athletes suggest that sleep is important for some aspects of physical performance. Depending on the sport discipline, extending nocturnal sleep or napping can improve cognitive function (e.g., reaction time and shooting accuracy), reduce daytime sleepiness, and enhance mood. Total sleep time required to achieve these benefits may vary and should be established on a case-by-case basis. Sleep hygiene in athletes can improve self-reported performance and facilitate training routine adjustment; however, studies often do not provide sufficient details about the implemented sleep hygiene protocols, and post-exercise recovery strategies are even less conclusive, highlighting the need for further research [81]. Sleep extension can improve sprint times, tennis serve accuracy, swim turn and kick stroke efficiency, swim sprint, basketball shooting accuracy, half-court and full-court sprints, and time to exhaustion. Enhanced cognitive function can include reaction times, psychomotor vigilance tasks, alertness, vigour, and mood. Altogether, the athletes with adequate sleep reported less fatigue and sleepiness [82].

The impact of sleep on muscle recovery is equally important for athletes. The precise mechanism of how sleep influences muscle recovery is not fully understood; however, there is evidence showing that sleep deprivation promotes a catabolic state in skeletal muscles via hormonal regulation. Sleep deprivation increases the level of cortisol and reduces levels of anabolic hormones: testosterone, GH, and insulin-like growth factor 1 (IGF-1), which can inhibit muscle protein synthesis [83]. Lamon et al. found an increase in plasma cortisol by 21% and a decrease in testosterone level by 24% after one night of complete sleep deprivation. Muscle protein synthesis rate was reduced by 18% [84]. Chase et al. evaluated the effects of sleep restriction after evening exercise and observed a performance decline in a 3-km cycling carried out the next morning compared to the control group with normal sleep. No difference was observed in muscle peak torque or soreness [85]. Dattilo et al. showed no influence of sleep deprivation on peak torque after eccentric exercise-induced muscle damage, hence no delay in muscle strength recovery. However, they observed increased levels of interleukin 6 (IL-6), IGF-1, cortisol, and cortisol to testosterone ratio [86]. Sleep extension may promote muscle regeneration by increasing IGF-1 concentration, while sleep deprivation can increase the levels of inflammatory agents, including TNF-α and IL-6, that can inhibit differentiation of myoblasts through the degradation of myoblast determination protein 1 (MyoD) and myogenin [87].

Military service often requires performing physically demanding tasks under insufficient sleep conditions. Running and cycling tests have shown that the aerobic performance of army soldiers is negatively affected by sleep loss. However, while submaximal performance can be impaired, maximal aerobic performance may not be significantly compromised by sleep reduction. The analysis of aerobic performance can be complicated when considering additional training or dietary interventions—limited energy intake reduces aerobic performance, yet it is not clear whether it has a greater impact than sleep restriction. Research on anaerobic capacity has shown conflicting results. Prolonged sleep restriction may reduce the mean power output of the upper body, yet peak power seems to be unchanged. No clear conclusion has been reached regarding the lower body. Studies on muscular strength have focused primarily on changes in handgrip strength with mixed results. For example, the muscle strength of the lower body can remain unaffected. Additionally, sleep loss can harm muscle endurance and performance of military-specific tasks [88].

Sleep deprivation negatively affects physical performance in athletes, although results remain inconclusive due to substantial variability in study protocols and experimental designs. Sleep restriction alters heart rate, ventilation, and respiratory frequency, while also leading to increased lactate accumulation and elevated cortisol levels. Concurrently, muscle glycogen concentration decreases, and the ability to restore glycogen stores is impaired. Furthermore, sleep restriction can reduce anaerobic power, tennis serving accuracy, isometric force, and average sprint times. Submaximal-effort sports are more vulnerable to the negative consequences of sleep loss than maximum-effort sports. Prolonged periods of sleep restriction are associated with increased sympathetic nervous system activity and reduced parasympathetic activity, which may promote an overreaching or overtraining state. In summary, these physiological disruptions may lead to a faster onset of exhaustion during physical effort [34]. Smithies et al. evaluated a group of “elite cognitive performers” (including pilots, air traffic controllers, surgeons, medical residents, emergency responders, process operators, and athletes) and observed that sleep loss impairs performance on monotonous, low-salience tasks. In contrast, the ability to perform more complex and cognitively stable tasks was relatively preserved. Sleep restriction may thus affect tasks requiring rapid strategic shifts or responsiveness to varied stimuli, potentially compromising adaptive decision-making in dynamic environments [89]. Total sleep deprivation, partial sleep restriction, and sleep fragmentation negatively affect emotional functioning in healthy individuals, as evidenced by reduced positive affect, increased anxiety symptoms, and diminished emotional reactivity [90].

### 5.2. Sleep Disorders (SD)

Sleep disorders are a heterogeneous group of conditions that disrupt sleep and are diagnosed, for example, using PSG and ACG. They include: insomnia (ID)—chronic difficulty falling asleep or maintaining sleep (up to 12.4% of the population) associated with psychological and genetic factors [91,92,93,94,95,96,97]; type 1 narcolepsy—autoimmune damage to orexin neurons causing excessive daytime sleepiness and cataplexy [98,99,100]; NREM parasomnias (sleepwalking, night terrors)—disturbances in the NREM-wake transition with motor and autonomic symptoms, partially genetically determined [101,102,103]; obstructive sleep apnoea (OSA)—collapse of the upper airways during sleep, often associated with obesity [104,105]; and restless legs syndrome (RLS)—an urge to move the legs that intensifies at rest, more common with iron deficiency [106,107,108]. Both short (<7 h) and long (>9 h) sleep increase the risk of mortality and cardiovascular diseases [109]; ID and OSA are strongly associated with autonomic dysfunction, metabolic disturbances, and inflammation [110]. Circadian rhythm disorders contribute to heart disease and worsen prognosis in infections [111,112], while ID more than doubles the risk of depression [113,114].

Sleep deficiency and poor sleep quality (≤7 h/night, fragmentation) are common in athletes and have numerous negative consequences: they impair physical and cognitive performance, prolong recovery, reduce immunity, increase injury risk (in youth, ≤8 h of sleep increases injury risk by 1.7 times); in adolescents, sleep deprivation strongly correlates with musculoskeletal injuries, while prolonged sleep disturbances increase injury risk; in elite and youth athletes, SD is common, often exceeding 50%, and is associated with poorer mental health. Sleep monitoring (e.g., interviews, actigraphy, behavioural therapy, sleep hygiene) should thus become part of injury prevention, training, and health optimisation [115]. 

### 5.3. Sleep Disturbances as a Risk of Other Diseases

Sleep disturbances in athletes have been increasingly recognised not only as performance-limiting factors but also as significant contributors to broader health risks. Inadequate sleep duration and poor sleep quality are independently associated with increased risks of musculoskeletal injuries, impaired cognitive functioning, mood disturbances, and delayed recovery processes in athletes [34]. High-quality sleep supports sustained focus and psychological stability during competition, potentially enhancing athletic performance. Additionally, it plays a crucial role in post-exercise recovery processes, thereby reducing the risk of injury and contributing to the prolongation of athletic careers [35]. Importantly, sleep-disordered breathing, particularly obstructive sleep apnoea, has been identified as a potential, underdiagnosed cause of sudden cardiac death during physical exertion in athletes, underscoring the cardiovascular risks associated with untreated sleep disorders [116]. Sleep deficiency promotes metabolic disturbances by affecting carbohydrate metabolism and the regulation of the expression level of appetite-related hormones, which may increase the risk of obesity and type 2 diabetes. This consequently reduces athletic performance and leads to long-term health complications [117]. Sleep disorders also negatively impact immune system function, decreasing immunity and prolonging recovery time following injuries or infections [118]. Collectively, chronic sleep disruption in athletes not only affects performance but promotes a multidimensional health threat that warrants routine screening, education, and targeted intervention as part of comprehensive sports medicine and training programs.

Total or partial sleep deprivation reduces physical performance, strength, power, and endurance, increases cortisol levels, lowers testosterone, and contributes to injuries and mood disturbances, especially in young athletes. High-quality sleep, sleep extension, and short naps improve recovery, reaction time, technical precision, and delay fatigue. Sleep disorders, jet lag, and unfavorable dietary habits intensify fatigue, weaken immunity and concentration, while a proper diet supports circadian rhythm and sleep quality. The negative consequences of sleep deprivation, circadian rhythm disruptions, and sleep disorders, leading to reduced performance and an increased risk of injuries, are presented in Table 2 [64,65,66,67,68,69,70,71,72,73,74,75,76,77,78,79,80,81,82,83,84,85,86,87,88,89,90,104,105,106,107,108,109,110,111,112,113,114,115].

## 6. Treatment of Sleep Disorders in Athletes

Pharmacological treatment of sleep disturbances in athletes entails numerous challenges and limitations, particularly in the context of anti-doping regulations governing elite sports. Several sedative and hypnotic medications are included in the World Anti-Doping Agency’s (WADA) list of prohibited substances or are subject to special usage restrictions, including benzodiazepines and certain “Z-drugs” such as zolpidem. Although effective in managing insomnia, these agents can impair cognitive functions, reaction time, and psychomotor performance. Consequently, at higher doses, they are classified as substances with proven abuse potential [119].

Zolpidem, while not explicitly prohibited by WADA, has raised concerns due to the risk of dependency and adverse side effects, especially in elite athletic populations. Cases of misuse, often in combination with alcohol or other sedatives, have led to increased scrutiny of its use in sports settings [120]. Athletes using such medications must be aware that improper or excessive intake may adversely affect not only sleep quality but also performance, recovery, and decision-making capabilities.

Cannabinoids represent another class of substances relevant to the management of sleep. While the non-psychoactive compound cannabidiol (CBD) is legal and may offer benefits for sleep, tetrahydrocannabinol (THC), the primary psychoactive component of cannabis, remains prohibited in sports competitions. Its detection in biological samples may result in disqualification, irrespective of the intent or mode of administration [119,121]. Furthermore, the dietary supplement market for CBD products is insufficiently regulated, leading to the potential for contamination with prohibited levels of THC.

Over-the-counter sleep-promoting supplements are also of concern as some products contain undeclared substances, including anabolic steroids or other psychoactive agents. This presents a serious risk of inadvertent doping violations amongst athletes who may unknowingly consume contaminated supplements [122].

Melatonin is a legal substance, frequently employed to support sleep in athletes, and a natural hormone that regulates the circadian rhythm. Melatonin supplementation has proven particularly effective in cases of delayed sleep phase syndrome, jet lag, and in athletes who train or compete during evening hours. It improves sleep quality without the adverse effects commonly associated with pharmacological hypnotics, such as daytime drowsiness, impaired balance, or cognitive decline [123]. Importantly, melatonin is not included in the WADA Prohibited List and is considered safe for short-term use. Melatonin has neuroprotective and antioxidant properties, which may further benefit recovery in athletic populations [124]. Amongst pharmacological agents with sleep-promoting properties, certain antidepressants, particularly those with sedative profiles, have been recognised for their efficacy in managing sleep disturbances. According to current anti-doping regulations, the therapeutic use of antidepressant medications in athletes is not prohibited, and such treatment does not require a Therapeutic Use Exemption (TUE) [119,125]. While the mechanisms through which these drugs modulate specific neurotransmitter systems in the central nervous system are well established, their effects on physical performance and athletic outcomes remain insufficiently investigated. Despite their widespread use in clinical practice, the implications of antidepressant therapy on exercise capacity and competitive performance in athletes have not been thoroughly investigated. This underscores the need for cautious and evidence-informed decision-making when considering these agents for sleep-related complaints in athletes [126].

Trazodone is a sedating antidepressant used for insomnia by blocking serotonin 5-HT_2_A, alpha-1 adrenergic, and histamine H_1_ receptors, thereby facilitating sleep onset and maintenance. Although extensively studied in general populations, its safety and efficacy in elite athletes remain poorly defined with no controlled trials directly assessing its impact on performance, recovery, or exercise physiology [127,128]. In primary insomniacs, a randomised, double-blind, placebo-controlled study administering 50 mg trazodone nightly for 7 days demonstrated improved sleep maintenance, increased slow-wave sleep, and reduced nocturnal awakenings, yet concurrently revealed next-day impairments in short-term memory, verbal learning, equilibrium, and arm muscle endurance [129,130]. These findings suggest potential psychomotor and alertness decrements following trazodone use. Pharmacokinetic studies in healthy subjects comparing controlled-release versus conventional trazodone confirmed its sedative profile, with reductions in subjective alertness and prolonged reaction times. However, these investigations did not involve athletes nor an evaluation of performance under physical exertion [131].

The effects of mirtazapine, an antidepressant, on sleep efficiency have been evaluated in healthy adults and general clinical populations. However, no studies have investigated the use of mirtazapine for insomnia or sleep disturbances in athlete populations, nor its effects on athletic performance or recovery metrics.

A controlled trial in young healthy volunteers (30 mg single dose) demonstrated that mirtazapine significantly improved sleep efficiency, increased slow-wave sleep, and reduced nocturnal awakenings, without notable effects on REM sleep. However, side effects such as next-day drowsiness, psychomotor impairment, and delayed reaction time have been consistently reported, raising concerns about its use in populations requiring high-level cognitive and physical function [132]. In general practice, the randomised DREAMING trial evaluated a low-dose of mirtazapine (7.5–15 mg/day) in patients with insomnia disorder. The study observed a statistically significant improvement in insomnia severity at 6 weeks compared to the group that received a placebo, yet the effects were not sustained at 12 weeks or at the 12-month follow-up. The side effects were more frequent with mirtazapine than those with the placebo, although the differences were not significant [133]. Although the sedative profile of mirtazapine poses potential risks for athletic performance, mainly impaired alertness and coordination, there is no conclusive data in the athletic context. Although mirtazapine is not prohibited by WADA [119], and its usage does not require a TUE [125], its application in athletes warrants careful clinical oversight.

Mianserin, a noradrenergic and specific serotonergic antidepressant (NaSSA), manifests sedative effects via antagonism of histamine H_1_ and α_2_-adrenergic receptors. These properties may facilitate sleep onset and extend sleep duration, yet the general sedative profile raises concerns in populations requiring high vigilance and physical performance. A randomised crossover study administering healthy volunteers mianserin (30–60 mg at night) over 16 days showed increased total sleep time and subjective restfulness. However, mild but considerable impairments in driving performance and psychomotor vigilance, especially early during treatment, persisted through day 16, indicating incomplete tolerance development [134]. Although mianserin is not listed on the WADA Prohibited List [119] and its use does not require a TUE [125], there are no studies evaluating its use in athlete populations, nor data on its impact on training metrics and recovery. Nevertheless, due to potential sedation, delayed reaction times, and psychomotor slowing, its use in athletes demands careful clinical oversight. Given the absence of specific evidence and the documented psychomotor effects, routine use of mianserin in sports medicine practice for sleep disorders is not advisable. Its administration may be considered only when behavioural and environmental interventions have failed, and using the lowest effective dose, with close monitoring of adverse effects and functional performance. 

From a clinical standpoint, pharmacological treatment in athletes of sleep disorders should be reserved for clearly defined medical indications and applied with caution. In such cases, the issuance of a TUE may be necessary. Chemical agents with a short half-life and low potential for dependency are preferred, and pharmacotherapy should be time-limited and closely monitored.

Teams should engage in systematic evaluation of sleep patterns and the daily routines of their athletes. Maintaining sleep and recovery logs can be a valuable tool for tracking changes in rest quality and assessing their impact on athletic performance. It is also essential to regularly review both home and travel schedules, with particular attention to their potential effects on sleep. Fostering a team culture that prioritises and supports healthy sleep practices should be a foundational element. Developing individualised sleep plans tailored to the physiological needs and circadian preferences of each athlete is recommended. In selected cases, the implementation of objective sleep monitoring tools may be beneficial. Additionally, athletes should be evaluated for common sleep disorders, such as insomnia or obstructive sleep apnoea [135].

Table 3 presents a summary of nutritional, behavioural, pharmacological strategies, and other factors shown to influence sleep parameters in athletes. The evidence is based on findings from clinical studies.

## 7. Recovery Through Sleep in Athletes

Adequate sleep is a fundamental determinant of recovery, cognitive functioning, and injury prevention in athletes, with specific requirements varying according to the physiological and psychological demands of different sport disciplines. Optimal sleep duration, structured napping protocols, and consistent sleep hygiene routines are complemented by tailored recovery strategies such as thermal interventions, psychological skills training, and relaxation techniques. Overall, the reviewed strategies consistently emphasize the need for sufficient nocturnal sleep (typically 8–10 h, with longer durations during intensive training), the use of short, early-afternoon naps to support recovery and alertness, and the maintenance of stable circadian-aligned routines. Core recommendations also include limiting evening exposure to stimulants and electronic devices, optimizing the sleep environment (cool, dark, and quiet), and incorporating structured recovery modalities such as foam rolling, cold- or heat-based therapies, and psychological techniques (relaxation, mindfulness, imagery, and breathing exercises). Collectively, these approaches aim to enhance sleep quality, accelerate recovery, and sustain both physical and cognitive performance across athletic populations [61,63,76,82,136,137,138,139,140,141,142,143,144,145,146,147,148,149,150]. Table 4 provides an overview of these evidence-based recommendations, highlighting both general and sport-specific considerations across endurance, strength, precision/technical, combat/fighting, and team sports.

## 8. Discussion

According to the International Olympic Committee (IOC), sleep is a major contributor to athletic performance and a fundamental feature of athlete mental health [150]. Adequate sleep is crucial for optimal recovery and subsequent performance in athletes [81]. Adequate sleep has been shown to have restorative effects on the immune system and endocrine system, facilitate nervous system recovery and the metabolic cost of wakefulness, and play a significant role in learning, memory, and synaptic plasticity, which can affect sports recovery, injury risk reduction, and performance [151]. Athletes who sleep adequately before competition are likely to benefit from the standpoint of peak performance [82]. For elite athletes who exercise at a high level, sleep is critical to overall health, general well-being and performance. Sleep is therefore as important as exercise routine and diet [82]. Research indicates that professional athletes should increase their sleep time. Athletes need 9–10 h of total sleep time to reach their full potential [82,152]. The need to extend sleep time stems from the physical and psychological demands of the sport [115]. However, numerous studies have also reported that athletes frequently fail to achieve the recommended 7–9 h of sleep because of both sport-related and external factors, thereby compromising the quality of their recovery [7]. Restoration of sleep and sleep extension are crucial for athletes to recover from exhaustion [153]. Improving sleep resulted in decreased fatigue and sleepiness [154]. Sleep extension (bank sleep) may bring potential benefits such as improving athletic performance and mood, reducing stress levels, and alleviating anxiety associated with participating in an important competition. By getting more sleep before an important competition, athletes can have confidence knowing that a poor night’s sleep the night before competition should not affect their performance [115]. The results of a study by Schartz et al. suggest that a minimum of 1 week of increased sleep duration leads to improvements on a range of performance measures among sleep-deprived athletes [155].

Sleep in athletes forms the foundation of regenerative and adaptive processes, serving as a key factor influencing physical performance, mental health, and injury prevention. The average sleep duration for adults is 7–8 h per day [1]; however, the regenerative needs of athletes, particularly those engaging in intensive training, are often greater. Differences in sleep duration and quality depend on gender, age, and training environment [1], indicating that sleep strategies should be individualised rather than universal.

In our narrative review, we aimed to comprehensively show how sleep affects the health and athletic results of athletes. We have demonstrated that sleep is no longer perceived as a passive element of recovery, but as an active and essential process that is fundamental to health, well-being, and achieving peak athletic performance. As Vitale et al. aptly conclude, the old adage “you snooze, you lose” should, in the world of sports, be: “you snooze (more), you win.” These authors emphasize that ignoring sleep can nullify the benefits of physical activity [82].

Athletes, especially at the collegiate level, are a group particularly vulnerable to sleep deprivation. Studies, such as those conducted by Mah and colleagues, clearly indicate that student-athletes often experience poor quality and insufficient amounts of sleep [156]. The reasons for this are complex and include intense training schedules, academic responsibilities, travel, and competition-related stress. The consequences of sleep deprivation for athletes are multidimensional and well-documented. They include impaired reaction time, decreased accuracy, reduced strength and endurance, and cognitive function deficits, such as decision-making [82]. Research confirms these observations in practice, showing, for example, that sleep restriction directly impairs maximal jump performance and joint coordination in elite athletes [157].

An additional factor is circadian rhythm disruption, often caused by traveling across time zones. This has a direct impact on performance, as observed in professional American football players [158]. The negative effects of travel, including fatigue and desynchronization of the biological clock, pose a serious challenge for athletes at every level of competition [63]. One of the most effective ways to counteract the negative effects of sleep deprivation is to consciously extend it. This strategy, known as sleep extension, has been the subject of numerous studies that have consistently shown its positive impact on athletic performance. A groundbreaking study by Mah et al. on collegiate basketball players [159] showed that extending sleep to a minimum of 10 h per night for 5–7 weeks brought tangible benefits. As also highlighted in the attached review article by Vitale et al., the basketball players showed improved sprint speed and a significant increase in shooting accuracy—for both free throws and 3-pointers—by over 9% [82]. Moreover, the athletes reported improved mood and overall well-being [159].

These benefits are not limited to basketball. Similar effects have been observed in other disciplines. Extended sleep improved reaction time off the blocks, turn efficiency, and overall well-being in swimmers [160]. In collegiate tennis players, an improvement in serve accuracy was noted. As in other team sports, additional sleep brought performance benefits [161]. Even in ultra-triathlons, sleep duration correlated with performance outcomes [162].

Currently, a newer concept is “sleep loading,” which involves deliberately extending sleep before a period of planned sleep deprivation. A study on professional baseball players showed that this strategy improves reaction time in visual search tasks and reduces perceived fatigue [163].

Scientific evidence clearly indicates that education about sleep is a key element of training programs. Educational initiatives, such as those conducted at Stanford University, yield positive, long-term effects [164]. The role of the entire staff—coaches, doctors, and physiotherapists—is crucial here, as they should collaborate on managing athletes’ sleep [165].

In their review, Vitale et al. provide practical tips on sleep hygiene that athletes can implement. These include maintaining regular waking and sleeping times, creating an optimal sleep environment (dark, quiet, cool room), avoiding stimulants (caffeine) and exposure to blue light from screens before bed, and introducing relaxing bedtime rituals [82].

Research shows that sleep deprivation in athletes leads to significant reductions in strength, power, endurance, and cognitive function, increasing the risk of injury, especially among younger athletes [5]. At the cellular level, sleep regulates immune and hormonal function as well as inflammatory cytokines, supporting post-exercise recovery and training adaptations [4]. Additionally, epigenetic mechanisms, such as DNA methylation and microRNAs, may modulate the physiological response to training [6]. High-quality and adequately long sleep thus acts as a natural “regeneration window”, promoting performance gains and reducing oxidative stress [7].

One of the major challenges for athletes is the frequent travel across time zones, which leads to circadian desynchronisation and jet lag symptoms. In practice, this manifests as sleep disturbances, decreased concentration, irritability, fatigue, and gastrointestinal issues, all of which impair both physical and mental performance [60,61]. The direction of travel also plays a critical role—east–west travel presents greater challenges for circadian adjustment, whereas westward travel may result in competitions coinciding with the biological night of the athlete [61]. The consequences of circadian disruption are discipline-specific: endurance athletes generally require more sleep than strength athletes, and sleep deficits can significantly impair training effectiveness and competition performance [156].

From a psychological perspective, sleep and stress are closely interconnected. Chronic stress increases cortisol secretion, weakens immunity, and impairs recovery, while sleep deficits negatively affect cognitive function and concentration [166,167]. Interdisciplinary strategies, including psychoeducation, lifestyle optimisation, physiological monitoring, and social support, can reduce allostatic load, while metacognitive techniques, pre-performance routines, and relaxation support can increase regulation and psychological resilience. The effectiveness of these methods, however, depends on the experience, sport-specific demands, and individualised program design [166,167].

Napping interventions aim to increase the amount of sleep obtained via total sleep duration or brief targeted naps [152]. Daytime sleep quantity (naps) can supplement insufficient nighttime sleep, as well as being beneficial for people who want a boost in alertness [115]. A systematic review conducted by Bonnar et al. found that naps taken later in the day and at an appropriately timed interval after previous exercise may positively influence an athlete’s mental disposition towards subsequent performance tasks. However, a 20-min nap may be too short to influence subsequent performance measures among partially sleep-deprived athletes. It may also disturb sleep on the subsequent night for some athletes [81].

It is worth noting that athletes often face challenges such as travel, late-night games, and pool lane hours, which make it difficult to maintain a regular sleep schedule during certain times. In this context, it is important for those working with athletes to seek alternative strategies aimed at ensuring healthy sleep in specific circumstances and conditions (such as choosing appropriate nap times or preparing an individual’s body clock for travel across time zones [81]. Recommendations for improving sleep also include taking into account individual chronotypes (evening vs. morning chronotype), i.e., avoiding training times early in the morning and late at night, depending on the individual preferences of athletes [115]. Chronotype is a significant variable in determining optimal times for training and competition. It may influence an athlete’s choice of sport or their success in it.

The importance of sleep education should also be emphasised, which consists of providing adequate informational support (knowledge about sleep hygiene) and instrumental support (specific tips to help develop healthy sleep habits). Promoting sleep information specific to the athlete’s sport, such as sleep needs, adjusting to training times and emphasising the impact of sleep on performance, is key to creating buy-in and behavioural change [115].

In addition to behavioural strategies aimed at developing healthy sleep habits, cognitive strategies are also used in working with athletes to reduce tension associated with anxiety and fear before competitions. Interventions involve identifying dysfunctional thoughts and seeking alternative solutions/scenarios, including progressive muscle relaxation and meditation. Battaglini et al. documented that progressive muscle relaxation effectively reduces cognitive anxiety and stress related to sports (competitions, trips, and overtraining), lowers heart rate, and promotes the athlete’s recovery [168]. Anderson et al. recognised the validity of using mindfulness as an intervention aimed at improving athlete health and performance [169]. Vveinhardt et al. determined that mindfulness practices are related to athletes’ higher performance and better psycho-emotional state. The psychological state indicators of kyokushin athletes who used mindfulness techniques (meditation, concentration) were better than those of athletes who did not use these techniques and were unaware of the benefits of mindfulness [170]. The benefits of meditation experience in mitigating the negative effects of mental fatigue have been documented in studies by Nien et al. [171]. Other studies have shown that after 20-week mindfulness meditation training interventions, male fencer athletes exhibited lower salivary cortisol concentrations and mental fatigue [172].

Sleep problems in athletes are related to both sport-specific factors (high training loads, travel, and stress associated with competition) and non-sport factors [115]. In the case of non-sport-related factors, it may be necessary to refer to various forms of psychological assistance (crisis intervention, psychoeducation or psychotherapy).

Pharmacological interventions for sleep disorders in athletes present a nuanced risk-benefit calculus. Hypnotics and sedative antidepressants may improve sleep, but often at the expense of cognitive and physical performance, while legal alternatives like melatonin may offer safer, if modest, benefits. These realities underscore the need for individualised strategies, close monitoring, and integration with behavioural and environmental interventions. The ultimate goal is to enhance recovery and sleep quality without undermining athletic performance or violating anti-doping rules. Pharmacological hypnotics, such as benzodiazepines or “Z-drugs” like zolpidem, can improve insomnia, yet the risks for athletes are notable. On one hand, they can facilitate faster sleep onset and longer duration; on the other hand, they impair reaction time, psychomotor function, and cognitive performance that are critical during training and competition [119,120]. Even when not explicitly prohibited by WADA, such as zolpidem, drugs carry dependency risks and may reduce next-day performance. From a practical standpoint, the potential gains in sleep may be offset by the declines in precision, coordination, or decision-making—core elements of athletic success.

Cannabinoids illustrate another grey area. While CBD appears safe and may support sleep, THC is a banned drug, and supplement contamination is common [119,121]. Therefore, athletes attempting a seemingly harmless intervention could face inadvertent doping violations. Over-the-counter sleep aids pose a similar dilemma, with undeclared psychoactive or anabolic substances potentially compromising the eligibility of the athlete to compete [122]. Thus, what may seem like a simple sleep solution can have far-reaching consequences for elite athletes.

Melatonin, by contrast, provides a compelling argument for use in sport. Its benefits for circadian adjustment, jet lag, and evening training schedules are well-documented, and its neuroprotective and antioxidant properties may support recovery [123,124]. Melatonin is also a legal drug under WADA rules, making it a lower-risk option in competitive sports. Yet, the question remains whether its effects on sleep translate into measurable performance gains in elite athletes—a topic still under investigation.

Sedative antidepressants—such as trazodone, mirtazapine, and mianserin—offer another layer of complexity. Studies in non-athletes show improved slow-wave sleep and reduced nocturnal awakenings [129,130,131,132,133,134]. However, side effects such as next-day drowsiness, psychomotor slowing, and delayed reaction times pose potential threats to athletic performance. The absence of athlete-specific trials means these medications are used empirically rather than based on rigorous evidence. While permitted under anti-doping regulations and not requiring a TUE [119,125], their sedative effects necessitate cautious application and monitoring, and individualised dosing. In some cases, the risks of using these sedative antidepressants may outweigh the benefits, particularly for athletes competing in sports requiring rapid reflexes and fine motor control.

Sleep in athletes is a dynamic process that not only enables rest but also actively supports regeneration, training adaptation, psychological resilience, and injury prevention. Understanding its importance in the context of intensive training, travel, and psychological stress allows for the design of personalised strategies to improve sleep quality, which in practice translates into optimised athletic performance and long-term health.

## 9. Strengths and Limitations of This Review

We provide a comprehensive and multidimensional review examining physiological (including molecular and hormonal) as well as dietary and pharmacological determinants affecting sleep in athletes. This allows us to present a wide spectrum of factors influencing sleep—from neurotransmitters (e.g., serotonin, GABA, orexin, melatonin) to metabolic and neuroendocrine mechanisms. Chronic partial sleep deprivation in athletes can lead to disturbances in carbohydrate metabolism and neuroendocrine function, which in turn result in adverse changes in appetite, food intake, and protein synthesis. These effects involve numerous signalling pathways and alter protein expression levels [173]. The in-depth review of these molecular pathways contributes to deepening our understanding of the physiological basis of the importance of sleep in physical performance.

We also highlight the link between sleep and the specificity of elite sport. We addressed not only the general restorative function of sleep (e.g., the role of slow-wave sleep in repair processes) but also factors unique to athletes. The specific demands related to training, travel, competition, as well as sex and stress level—factors that significantly modulate sleep in athletes—are highlighted. Sport-specific factors (relating to training, travel, and competition) and non-sport factors (e.g., females, stress, and anxiety) can influence sleep in athletes [115]. This multidimensional perspective is consistent with the current approach to optimising sleep for athletes. For example, the International Olympic Committee (IOC) and National Collegiate Athletic Association (NCAA) recognise healthy sleep as a fundamental component of health and performance and recommend individualised strategies for its improvement [34]. We therefore demonstrate alignment with the latest consensus statements and scientific findings.

It is worth noting the limitations of this review, including the generalised treatment of athletes as the homogeneous group. This simplification overlooks the fact that different sports impose different sleep demands. For example, studies have shown that American football (National Football League) players are more likely to experience obstructive sleep apnoea than other athletes or the general population. Similarly, endurance athletes may respond differently to sleep loss compared to sprinters or weightlifters. Such uniform categorisation diminishes nuances related to sport disciplines or training level [34].

Another drawback lies in the selective choice of factors affecting sleep. The review focuses on classical determinants (such as dietary macronutrients and common supplements) and well-established neurochemical mechanisms, but omits less obvious yet potentially important aspects, such as a high-fibre diet—typical of the Mediterranean dietary pattern—investigated for its impact on sleep, associated with a significant increase in deep sleep at the expense of light sleep [174]. Excluding those diets, and despite evidence that fruit-, vegetable-, and whole grain–rich diets can improve sleep quality, is an example of selective variable inclusion. It is also worth emphasising that a high intake of fibre, particularly from whole grains, vegetables, and fruits, substantially increases meal volume and enhances satiety. Elevated energy requirements, typical of endurance sports, may lead to an unintentional reduction in total energy intake. Moreover, diets rich in fibre are characterised by low energy density, which further complicates meeting the caloric needs of female athletes. As a result, there is an increased risk of low energy availability (LEA), a condition associated with hormonal disturbances, reduced bone mineral density, and impaired recovery capacity. Although dietary fibre provides health benefits, excessive intake in female endurance athletes may have detrimental effects and therefore should be carefully managed when planning nutrition strategies [175]. A study of 364 endurance athletes identifying strategies used to alleviate exercise-associated gastrointestinal symptoms recommended pre-competition dietary modifications, such as reducing fibre and fat intake, adhering to a low fermentable oligosaccharides, monosaccharides, and polyols (FODMAP) diet, and implementing gut training. The most effective strategies for athletes were temporary fibre restriction, elimination of fermentable carbohydrates, and gradual adaptation of the gastrointestinal tract to carbohydrate intake during exercise. These practices are largely empirical and require further interventional studies for verification, highlighting the need for the development of standardised nutritional guidelines for endurance athletes [176,177].

In this review, studies were limited to molecular pathways, while mechanisms of the circadian clock or the detailed neuroendocrine effects of stress and neurotransmitters regulating sleep were not explored.

Significant challenges also arise in evaluating less tangible factors, such as psychological stress and mental load. Although these can be risk factors, their quantitative assessment remains difficult.

This review is constrained by the considerable heterogeneity of the included clinical studies. Participant characteristics varied widely in terms of age, sex, training status, and competitive level, ranging from recreationally active individuals to elite professional athletes. Sample sizes differed markedly between studies, from very small cohorts to larger observational groups, which complicates cross-study comparisons and limits the generalizability of the findings. Additionally, study designs were diverse, including randomized controlled trials, crossover studies, prospective cohorts, and cross-sectional surveys, often with differing intervention protocols, outcome measures, and assessment tools. Such variability reduces the ability to draw definitive conclusions regarding the effects of dietary factors on sleep in athletic populations. Furthermore, few studies investigated the interactions between meal timing, training intensity, and other environmental or lifestyle factors. Moreover, sleep measurement methods in sports research have limitations. Although ACG devices or screening questionnaires are often used, they are less accurate in assessing true sleep architecture and cannot replace the advantages of PSG. It is worth noting that these methods were not comprehensively reviewed, which may limit the interpretation of our findings [115]. Differences in metabolic demands and training loads across sports were not consistently considered, further limiting the applicability of results across athletic disciplines. These limitations underscore the need for more standardized, well-powered, and methodologically rigorous research in this field.

Finally, we highlight research directions and outline many questions regarding the impact of sport on sleep that require further verification and in-depth analysis. The lack of female participants in experimental studies and the limited experimental control in researching sleep in athletes remain gaps in the literature. There is also a need to validate and refine the tools used to assess sleep, ensuring they are suited to athletic populations. From a practical standpoint, future studies could examine the effects of napping and sleep extension (“banking sleep”) on performance [115], as well as the role of dietary interventions. Nutritional interventions can therefore modulate the sleep–wake cycle and are considered promising interventions, suggesting that further experiments in this area may yield new strategies to support athlete recovery.

## 10. Summary

The management of sleep disturbances in athletes should prioritise non-pharmacological interventions, which have proven to be both effective and safe. These approaches, including cognitive behavioural therapy for insomnia (CBT-I), can significantly improve sleep quality, mental health, recovery, and athletic performance. While pharmacological treatment may be necessary in selected cases, it must be approached with caution, requiring not only medical expertise but also thorough consideration of anti-doping regulations and potential side effects. Consequently, pharmacological strategies should be reserved for situations in which behavioural interventions have failed, and only under the close supervision of qualified healthcare professionals. Ultimately, safeguarding the health and safety of the athlete, and upholding the integrity of competitive sport must remain central to all therapeutic decisions.

Treatment strategies should incorporate consistent lifestyle and environmental modifications. Regular exposure to daylight, caffeine and alcohol restrictions, sleep hygiene optimisation, and the implementation of calming pre-sleep routines can enhance sleep quality and duration. Maintaining consistent sleep–wake schedules, integrating short restorative naps, and minimising evening exposure to electronic devices can also contribute to recovery. Team-based approaches, including routine assessment of sleep habits, travel schedules, and individualised sleep plans, are essential. Objective monitoring and early identification of disorders such as insomnia or obstructive sleep apnoea should form a core component of comprehensive athlete care (Figure 7).

## 11. Conclusions

Optimal sleep duration and quality represent a critical “regenerative window”, essential for enhancing athletic performance and safeguarding physiological resilience. Ensuring adequate sleep among athletes can be effectively achieved through the non-pharmacological interventions outlined in this review. This review uniquely substantiates this by examining the molecular consequences of sleep deprivation, such as the activation of pro-apoptotic gene pathways (BAX, CCAR2) and the disruption of circadian regulators (BMAL1). Furthermore, the work presents a novel, practical analysis of pharmacological options, critically evaluating specific medications in the context of their impact on athletic performance and compliance with anti-doping regulations. Such an integrated approach, linking cellular mechanisms with clinical and nutritional strategies, provides a comprehensive understanding of how to optimise this critical regenerative window.

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
