# Peer review of "Sleep and Athletic Performance: A Multidimensional Review of Physiological and Molecular Mechanisms"

_jcm, 2025, doi:10.3390/jcm14217606_

Round 1

Reviewer 1 Report

Comments and Suggestions for Authors

Sleep is a fundamental component of the life cycle of all living beings. Accordingly, the study of sleep quality and duration in athletes is highly relevant and important. One of the undeniable strengths of the manuscript is that the authors present a broad and multidimensional review covering physiological, molecular, and applied aspects, as well as practical recommendations for athletes. The manuscript is well-structured, supported by a large number of references, including recent publications from 2020–2024, and can be of value to specialists in sports medicine. Nevertheless, several revisions are required to further improve the quality of the manuscript.

First, the methodology of the literature search is not described with sufficient transparency. Although PubMed and Cochrane are mentioned, clear inclusion and exclusion criteria are lacking. For greater rigor, the authors should consider providing a PRISMA flow diagram or at least a table outlining the stages of the screening process. Second, the review is at times more descriptive than analytical, with a tendency toward compilation of data rather than critical evaluation. Structuring more of the data into tables with references would improve readability. It would also be beneficial to emphasize the limitations of existing studies, such as small sample sizes, heterogeneous populations, and variability in sleep assessment methods. Third, although the paper is presented as focusing on athletes, a significant portion of the literature discussed relates to the general population (e.g., studies on Alzheimer’s disease or β-amyloid). While interesting, these findings do not directly align with the stated objective of the review; a stronger emphasis on athlete-specific data would be more appropriate. Fourth, the practical recommendations, though useful, remain rather general (e.g., 7–9 hours of sleep, avoiding caffeine in the evening). The paper could be strengthened by providing more tailored guidance depending on the type of sport—for instance, endurance, strength, or team disciplines.

The English language is generally adequate, but some sentences are overly complex and lengthy, which reduces readability. A round of professional language editing would improve the overall flow of the manuscript. Finally, an additional summary table linking sleep-related factors to specific performance outcomes in athletes would be a valuable addition to enhance accessibility for readers.

Author Response

Point-by-point response to the reviewer’s comments

First, the methodology of the literature search is not described with sufficient transparency. Although PubMed and Cochrane are mentioned, clear inclusion and exclusion criteria are lacking. For greater rigor, the authors should consider providing a PRISMA flow diagram or at least a table outlining the stages of the screening process.

Response:

We sincerely thank you for this valuable comment. We would like to emphasise that our manuscript is a narrative review, and therefore, it does not strictly adhere to the PRISMA guidelines, which are intended for systematic reviews. Instead, our work follows the general principles recommended for narrative reviews, such as those outlined in the SANRA guidelines.

Although our work is a narrative review and therefore does not fully adhere to PRISMA guidelines, we agree that providing a flow diagram improves transparency. Accordingly, we have prepared and added a PRISMA-style flowchart illustrating the process of literature identification, screening, and selection. The diagram is now included in the revised version of the manuscript.

Second, the review is at times more descriptive than analytical, with a tendency toward compilation of data rather than critical evaluation. Structuring more of the data into tables with references would improve readability. It would also be beneficial to emphasize the limitations of existing studies, such as small sample sizes, heterogeneous populations, and variability in sleep assessment methods. 

 Response:

We would like to thank the Reviewer for this valuable comment. In response, we have restructured some of the descriptive parts of the manuscript and added several summary tables to improve readability and provide a more analytical overview. Specifically:

Table 2. Relationship between sleep factors and athletic performance in athletes

Table 3. Evidence-based strategies and factors influencing sleep parameters in athletes

Table 4. Sleep and recovery in athletes: strategies to enhance sleep quality and performance

As suggested, the added tables are highlighted in yellow in the revised manuscript for easier identification.

Third, although the paper is presented as focusing on athletes, a significant portion of the literature discussed relates to the general population (e.g., studies on Alzheimer’s disease or β-amyloid). While interesting, these findings do not directly align with the stated objective of the review; a stronger emphasis on athlete-specific data would be more appropriate. 

Response:

We appreciate the Reviewer’s observation. We would like to clarify that the sections referring to the general population were included primarily to illustrate underlying physiological processes relevant to sleep and cognition, which also have implications for athletes. Nonetheless, we acknowledge the Reviewer’s concern and, in the revised version, we have placed a stronger emphasis on athlete-specific data, while keeping the general population findings only as supportive background. The corresponding additions and modifications in the text are highlighted in yellow for easier identification.

Fourth, the practical recommendations, though useful, remain rather general (e.g., 7–9 hours of sleep, avoiding caffeine in the evening). The paper could be strengthened by providing more tailored guidance depending on the type of sport—for instance, endurance, strength, or team disciplines.

Response:

We thank the Reviewer for this valuable suggestion. In the revised manuscript, we have expanded the section on practical recommendations by including more tailored guidance depending on the type of sport (e.g., endurance, strength, and team disciplines). While we retained general recommendations as a foundation, we added sport-specific considerations to make the guidance more applicable for athletes in different contexts. To further support this section, we have also included Table 4. Sleep and recovery in athletes: strategies to enhance sleep quality and performance. The corresponding additions and the new table are highlighted in yellow in the revised text for easier identification.

A round of professional language editing would improve the overall flow of the manuscript.

Response:

We thank the Reviewer for this remark. We would like to inform that the manuscript has been professionally proofread, and we obtained a certificate of language editing. This certificate was included in the documentation submitted to the Publisher during the article submission process.

An additional summary table linking sleep-related factors to specific performance outcomes in athletes would be a valuable addition to enhance accessibility for readers.

Response:

We appreciate this insightful suggestion. We agree that an additional summary table directly linking sleep-related factors to specific performance outcomes in athletes would be highly valuable. However, such a comprehensive analysis would considerably increase the length of the current manuscript and could compromise its readability, as it would go beyond the intended scope of this review. Nevertheless, we find this comment extremely valuable and will consider preparing a separate manuscript focused exclusively on the relationships between sleep-related factors and specific athletic performance outcomes.

Reviewer 2 Report

Comments and Suggestions for Authors

“Sleep and Athletic Performance: A Multidimensional Review of Physiological, Molecular, and Behavioural Mechanisms”(jcm-3869734)

This review aimed to provide a comprehensive view about the relationships between sleep and athletic performance. The authors conclude that Optimal sleep duration and quality represent a critical "regenerative window", essential for enhancing athletic performance and safeguarding physiological resilience. Ensuring adequate sleep among athletes can be effectively achieved through educational, behavioural, and nutritional interventions. Overall, this topic is interesting while was extensively explored and many related reviews have already been published. Some concerns appeared after reading the whole manuscript.

  1. The literature review part is far from satisfactory.

Some important review papers need to be reviewed and discussed which I mentioned above. What are the really novelties of current review need to be reconsidered.

  1. Since “anxiety, stress” were used as search keywords, then why did you not include “depression”? moreover, why did you include these mental health-related keywords as they seem did not directly relate to the topic (Sleep and Athletic Performance) of this manuscript.

  1. It would be very strange that one paragraph only includes only one sentence, such as “Sleep architecture also provides optimal conditions for hormonal, neurological, and anabolic processes [4].”

  1. Line 87, “A systematic review” is contradict with the type of this manuscript “Narrative review” in line 1. Is this manuscript a systematic review or narrative review? Which report guideline did you follow?

  1. The inclusion and exclusion criteria of literature should be more detailed in the formal manuscript.

  1. A figure of literature screen as included in PRISMA-2020 would be helpful to the readers to get better understanding of how did you pick the literature.

  1. The structure of this manuscript really confuses me. In my opinion, “3. The physiology of sleep”,“4. Methods and tools for assessing sleep quality and duration” were not necessary since they are not related to athletes. Moreover, some parts of both sections are neither comprehensive nor quite accurate. Instead, how sleep and athletic performance interacted should be describes first. And then why did this happen, and lastly the available management strategies.

  1. Although you mentioned “nutritional interventions”, however, little information was provided in the formal context and “nutritional interventions” was missing in figure 7.

  1. There are three paragraphs in the abstract, which is not the common routine.

  1. Are you sure Figure 2 correctly depicts the EEG patterns for each sleep stage?

Author Response

Point-by-point response to the reviewer’s comments

  1. The literature review part is far from satisfactory. Some important review papers need to be reviewed and discussed which I mentioned above. What are the really novelties of current review need to be reconsidered.

Response:

We sincerely thank you for this valuable comment. Indeed, numerous “related reviews” addressing the impact of sleep on athletic performance have been published. For example, the works of Vitale et al., as well as several papers cited in the comprehensive reviews by Mah et al. and others, serve as notable references. We have now included a discussion of these studies in our review, with particular attention to the aspects of “novelty” emphasised by individual authors. The added sections in the Discussion are highlighted in yellow.

We also appreciate your suggestion to reconsider what constitutes the true novelty of our review. In response, we clarified in the Conclusions that the unique contribution of our work lies in offering a multidimensional synthesis of the subject, combining the most recent molecular-level findings with their practical implications for sports medicine. The added sections in the Discussion are highlighted in yellow.

2. Since “anxiety, stress” were used as search keywords, then why did you not include “depression” moreover, why did you include these mental health-related keywords as they seem did not directly relate to the topic (Sleep and Athletic Performance) of this manuscript.

Response:

We thank you very much for this valuable remark. Following your suggestion, we have carefully revised the keywords of our manuscript, removing those that were not directly related to the central topic of our review (sleep and athletic performance).

However, within the manuscript, we intentionally referred to the psychological aspects of athletes’ well-being, including the impact of stress and anxiety, as these factors may influence both sleep quality and performance outcomes. We also briefly addressed pharmacological approaches, such as antidepressant therapy, given their potential relevance in sports medicine practice when sleep disturbances coexist with mental health challenges.

3. It would be very strange that one paragraph only includes only one sentence, such as “Sleep architecture also provides optimal conditions for hormonal, neurological, and anabolic processes [4].”

Response:

We thank you for this insightful comment. We have expanded the information on sleep architecture not only by elaborating on the sentence “Sleep architecture also provides optimal conditions for hormonal, neurological, and anabolic processes [4].” in the Introduction, but also by adding a more detailed section in The Physiology of Sleep. The newly added content has been highlighted in yellow in the manuscript for clarity.

4. Line 87, “A systematic review” is contradict with the type of this manuscript “Narrative review” in line 1. Is this manuscript a systematic review or narrative review Which report guideline did you follow?

Response:

We sincerely thank you for pointing out this important inconsistency. Our manuscript is indeed a narrative review. The reference to a “systematic review” was an oversight on our part. While our work includes systematic elements, such as a structured search of the scientific literature on the topic, the selective focus on studies that provided the most comprehensive insights into particular aspects—while omitting other similar papers, including some large meta-analyses—clearly positions this manuscript as a narrative rather than a systematic review. We have revised the text accordingly to ensure consistent terminology throughout the manuscript, explicitly presenting our work as a narrative review.

5. The inclusion and exclusion criteria of literature should be more detailed in the formal manuscript.

Response:

We thank you for raising this point. We would like to emphasise that our manuscript is a narrative review, and therefore, it does not strictly adhere to the PRISMA guidelines, which are intended for systematic reviews. Instead, our work follows the general principles recommended for narrative reviews, such as those outlined in the SANRA guidelines.

Nevertheless, to enhance methodological clarity, we have included a detailed description of our literature search strategy in the Methods section. In brief, we performed stepwise searches in the PubMed/Medline and Cochrane databases using a predefined set of English keywords and their combinations. Approximately 7,700 publications were initially identified. After title and abstract screening, 178 papers were selected for evaluation, and additional references were included by screening the bibliographies of key works. We excluded studies focused exclusively on psychological interventions, research involving children or older adults, and papers not directly related to sleep in athletes. The final selection covered literature on sleep physiology, the bidirectional relationship between sleep and exercise, diet and stress, sleep disorders, pharmacological and non-pharmacological treatments, and sleep assessment tools.

6. A figure of literature screen as included in PRISMA-2020 would be helpful to the readers to get better understanding of how did you pick the literature.

Response:

We thank you for this valuable suggestion. Although our work is a narrative review and therefore does not fully adhere to PRISMA guidelines, we agree that providing a flow diagram improves transparency. Accordingly, we have prepared and added a PRISMA-style flowchart illustrating the process of literature identification, screening, and selection. The diagram is now included in the revised version of the manuscript.

7. The structure of this manuscript really confuses me. In my opinion, “3. The physiology of sleep”“4. Methods and tools for assessing sleep quality and duration” were not necessary since they are not related to athletes. Moreover, some parts of both sections are neither comprehensive nor quite accurate. Instead, how sleep and athletic performance interacted should be describes first. And then why did this happen, and lastly the available management strategies.

Response:

We sincerely thank the Reviewer for this valuable comment regarding the structure of our manuscript. We agree that the relationship between sleep and athletic performance is central to the paper and should be introduced earlier. In response, we have revised the structure and added a new section at the very beginning of “3. The physiology of sleep”, which now describes in detail how sleep and athletic performance interact. This part precedes the more general physiological background and provides readers with a direct link between sleep and sports outcomes. The added sections in the Discussion are highlighted in yellow.

8. Although you mentioned “nutritional interventions”, however, little information was provided in the formal context and “nutritional interventions” was missing in figure 7.

Response:

We thank the reviewer for this valuable comment. We have expanded the section on nutritional interventions and have updated Figure 7 to include these interventions. The newly added content has been highlighted in yellow for clarity.

9. There are three paragraphs in the abstract, which is not the common routine.

Response:

We thank the reviewer for this observation. The abstract has been revised: paragraph breaks have been removed, and the text has been shortened by omitting methodological details to align with common formatting practices.

10. Are you sure Figure 2 correctly depicts the EEG patterns for each sleep stage

Response:

We thank the reviewer for this comment. Figure 2 has been revised to more accurately depict the EEG patterns for each sleep stage.